# The Epigenesis of Salivary Glands Carcinoma: From Field Cancerization to Carcinogenesis

**DOI:** 10.3390/cancers15072111

**Published:** 2023-03-31

**Authors:** Norhafiza Mat Lazim, Anam Yousaf, Mai Abdel Haleem Abusalah, Sarina Sulong, Zul Izhar Mohd Ismail, Rohimah Mohamud, Hashem A. Abu-Harirah, Tareq Nayef AlRamadneh, Rosline Hassan, Baharudin Abdullah

**Affiliations:** 1Department of Otorhinolaryngology-Head and Neck Surgery, School of Medical Sciences, Universiti Sains Malaysia, Health Campus, Kubang Kerian 16150, Kelantan, Malaysia; 2Hospital USM, Health Campus, Universiti Sains Malaysia, Kubang Kerian 16150, Kelantan, Malaysia; 3Department of Molecular Pathology Laboratory, Pakistan Kidney and Liver Institute and Research Centre, Lahore 54000, Pakistan; 4Department of Medical Laboratory Sciences, Faculty of Allied Medical Sciences, Zarqa University, Al-Zarqa 13132, Jordan; 5Department of Medical Microbiology and Parasitology, School of Medical Sciences, Universiti Sains Malaysia, Kota Bharu 16150, Kelantan, Malaysia; 6Department of Immunology, School of Medical Sciences, Health Campus, Universiti Sains Malaysia, Kubang Kerian 16150, Kelantan, Malaysia; 7Human Genome Centre, School of Medical Sciences, Health Campus, Universiti Sains Malaysia, Kubang Kerian 16150, Kelantan, Malaysia; 8Department of Anatomy, School of Medical Sciences, Health Campus, Universiti Sains Malaysia, Kubang Kerian 16150, Kelantan, Malaysia; 9Department of Haematology, School of Medical Sciences, Health Campus, Universiti Sains Malaysia, Kubang Kerian 16150, Kelantan, Malaysia

**Keywords:** Salivary gland cancers, epigenetic modifications, DNA methylation, noncoding RNAs, histone modifications

## Abstract

**Simple Summary:**

Salivary glands carcinoma are prevalent in head and neck surgical oncology practice. The treatment is challenging due to late diagnosis and high recurrence risk. The epigenetic event is one of the most important etiologies known for this cancer. At present, numerous pathways and epigenetic alteration has been identified. This epigenetic event may serve as a novel avenue for the development of effective diagnostic and therapeutic agents at near future. This will enhance the management of this type of cancer. In this review, we discuss the main epigenetic events in salivary gland carcinogenesis and highlight their roles in the prognostication and refined management of this cancer.

**Abstract:**

Salivary gland carcinomas (SGCs) are a diverse collection of malignant tumors with marked differences in biological activity, clinical presentation and microscopic appearance. Although the etiology is varied, secondary radiation, oncogenic viruses as well as chromosomal rearrangements have all been linked to the formation of SGCs. Epigenetic modifications may also contribute to the genesis and progression of SGCs. Epigenetic modifications are any heritable changes in gene expression that are not caused by changes in DNA sequence. It is now widely accepted that epigenetics plays an important role in SGCs development. A basic epigenetic process that has been linked to a variety of pathological as well as physiological conditions including cancer formation, is DNA methylation. Transcriptional repression is caused by CpG islands hypermethylation at gene promoters, whereas hypomethylation causes overexpression of a gene. Epigenetic changes in SGCs have been identified, and they have been linked to the genesis, progression as well as prognosis of these neoplasms. Thus, we conduct a thorough evaluation of the currently known evidence on the involvement of epigenetic processes in SGCs.

## 1. Introduction

The embryonic development of the tubulo-acinar exocrine organ known as the salivary gland begins between week 6 and week 8 of intrauterine life. Submandibular and sublingual glands originate in the embryonic endoderm, while the parotid gland is thought to develop from the oral ectoderm [1]. Salivary glands have a two-tiered structure with luminal (acinar and ductal) and abluminal (myoepithelial and basal) cell layers. Rapid entry into the cell cycle makes these cells vulnerable to neoplastic transformation [2]. Salivary gland carcinomas (SGCs) are uncommon compared to the other carcinoma types but are common in the context of head and neck tumours [3,4]. Salivary gland carcinomas (SGCs) account for between 3–6% of all head and neck malignancies. The parotid gland is the most commonly involved, especially by benign type followed by the submandibular gland and the minor salivary glands. Among the malignant histological subtypes are mucoepidermoid carcinoma (MEC), carcinoma ex pleomorphic adenoma, intraductal carcinoma, acinic cell carcinoma, adenoid cystic carcinoma (ACC), and carcinosarcoma [5,6,7]. Mucoepidermoid carcinoma is further classified into a low-grade and high-grade tumor where the treatment approaches are significantly differed. It is challenging to get earlier diagnosis of these SGCs and deliver adequate treatment due to existing high histological heterogeneity.

Salivary gland carcinomas (SGCs) are exceptionally rare, hence very little is known about their etiology. A few studies have reported that alcohol consumption, tobacco use, diet high in animal fat and low in vegetables, and heavy cell phone use are associated with an increased risk of SGCs [8,9]. Radiation exposure (such as radiotherapy to the head and neck) and certain occupational exposures (such as silica dust, nickel alloy dust, asbestos, and rubber products manufacturing and mining) have also been implicated [9]. A history of cancer [10] and perhaps exposure to the human papillomavirus [11], Epstein Barr virus [9], and HIV [12] have also been identified (Figure 1).

Epigenetic and genetic changes have been proposed as etiological variables, but there are yet few research investigating its function in SGT (Figure 2) [1,3,13]. Epigenetic events can take the form of DNA methylation, alterations in the expression of non-coding RNAs such as microRNAs (miRNAs), or abnormalities in the structural modification of histones [14,15,16]. Several cancers, including SGCs, develop and progress due to epigenetic alterations that cause considerable changes in gene expression [3]. In addition, significant genetic alterations have been documented in all SGCs, and these alterations can be grouped according to their role in prediction, diagnosis and prognosis [13].

The development of molecular biology techniques has allowed for a better understanding of the histogenic, morphogenic, and genetic mechanisms that underlie different SGCs. This has resulted in more efficient methods of diagnosis and, consequently, more effective therapy techniques [1]. Salivary gland diseases are notoriously difficult for pathologists and clinicians because of their stochastic nature, which necessitates a constantly evolving classification system. The purpose of the current review article is to provide a comprehensive overview of the most up-to-date information regarding the role of epigenetic alterations as predictive, diagnostic biomarkers and therapeutic targets in the management of SGCs.

## 2. Epigenetics Mechanisms

Epigenetics is a broad word that refers to molecular mechanisms that affect gene expression without altering the DNA base sequence. Transcription regulators, epigenetic writers, gene imprinting, histone modification as well as DNA methylation are important epigenetic processes implicated in gene expression alterations (Figure 2, Table 1) [17]. DNA methylation involves transformation of methylated cytosine by treatment with sodium bisulfite, into thymine and two distinct probes which used to target each site of CpG [18]. Mechanism of histone modifications involves chemical post-translational modifications (PTMs) such as sumoylation, ubiquitylation, acetylation, phosphorylation as well as methylation, to the histone proteins, that causes chromatin structure to change or attract histone modifiers [19]. Another epigenetic mechanism is genomic imprinting that impacts a small group of genes, resulting in monoallelic expression of genes which is parental specific origin in manner. Gene expression as well as genomic region compaction are controlled by epigenetic alterations, which are produced by specific enzymes called as “writers” and eventually identified by the effector proteins called as “readers” and removed by erasers, all of which together contribute to the regulation of gene transcription, and abnormalities can result in tumor formation as well as development [20]. Additionally, the creation of a different research known as nutrigenomic results from epigenetic regulation via diverse nutritional substances [21,22].

### 2.1. DNA Methylation

In vertebrates, 5-methylcytidine is the product of DNA methylation occurring at position 5C of cytosine residues, which arises mainly inside of CpG dinucleotides. Methylation of Non-CpG occurs (particularly CpHpH as well as CpNpG methylation, where H ¼ A, T, C). As intermediates in DNA demethylation mechanism, additional configurations of cytosine involve 5-carboxylcytosine, 5-formylcytosine as well as 5-hydroxymethylcytosine [54,55,56]. Nearly 70–80% of CpG dinucleotides have been reported to be methylated in mammalian genomes. There is an existence of CpG islands (CGIs), which are referred to as regions with CpG rich sequences accompanied by reduced degree of DNA methylation [57,58]. DNA methylation is a primary pathway involved in epigenetic gene suppression. During differentiation, DNA methylation (de novo) targets the germline specific genes as well as promoters of stem cells. MethylCpG-binding proteins are also involved in DNA methylation pathway. In succession, these proteins interact with additional proteins, resulting in silencing alterations to adjacent histones. The overall process of coordinating silencing histone marks as well as DNA methylation induces gene suppression as well as chromatin compaction [59,60,61].

#### 2.1.1. CpG Islands

Human genome account for just 0.7% of CGIs but account for 7% of CpG dinucleotides. At gene promoters, CGIs are often more abundant. Approximately, 60% of all gene promoters of mammals reported to be CpG rich. Unmethylated CGIs are found in open locations of DNA having minimal nucleosome occupancy [61,62]. CGIs promote euchromatin, or relaxed chromatin structure that favors active transcription and increases the availability of additional elements of basal transcription mechanism as well as RNA polymerase II to an initiation site of transcription. Most CGI promoters have variable transcription start sites and lack TATA boxes. Variable transcription initiation sites are possessed by most of the CGI promoters with absence of TATA boxes. Consequently, TATA-binding proteins are inducted by SP1 (referred as transcriptions factors having CpG in their identification region), to the promoters devoid of TATA boxes [21,61,62].

Transcription from CGI promoters aid in nonproductive, bidirectional cycles of initiation, and premature termination. The regulatory signals required for the progression of this nonproductive state to productive is not understood properly till date. Similarly directional synthesis of full-length transcripts is not properly characterized till date. The mechanisms associated with CpG islands free of methylation appear to involve binding of transcription factors and other transcriptional machinery or the act of transcription itself. CpG islands may be hypermethylated for silencing specific genes during cellular differentiation, genomic imprinting, and X chromosome inactivation [21].

Transcription via CGI promoters assist in premature termination as well as bidirectional and inefficient initiation cycles. Regulatory signals necessary for transition from this inefficient to efficient state are still poorly understood. Furthermore, targeted formation of transcripts (full-length) has yet to be thoroughly studied. Transcriptional machinery, transcription factor binding as well as process of transcription itself are the mechanisms linked with methylation free CGIs. During X chromosome inactivation, genomic imprinting as well as cellular differentiation, CGIs may be hypermethylated to silence certain genes [61,62].

#### 2.1.2. DNA Methylases

DNA methyltransferases are enzymes that catalyze DNA methylation (DNMT) [63]. DNMT3B as well as DNMT3A are involved in de novo methylation and are targeted to specific genomic areas via histone changes [53]. Protein Np95 detects hemimethylated DNA and sends DNMT1 towards replication fork (RF) to modify DNA methylation patterns, during DNA replication [64].

### 2.2. Histone Modifications

Covalent PTMs to a histone protein are referred to as histone modifications that can include sumoylation, ubiquitylation, acetylation, phosphorylation as well as methylation. Gene expression is influenced by PTMs to histones which results in recruitment of histone modifiers as well as altered chromatin structure. Histone variations that make up histone modification as well as nucleosome, are used to regulate epigenetic genes. Conventional nucleosomes incorporate proteins like, H2A, H2B, H3 as well as H4 and are octamers. There are various histone variations that differ by massive insertions or by limited number of amino acids [65]. These histone variations are often discovered at particular sites inside of chromatin or are utilized to demarcate the borders between euchromatin as well as heterochromatin areas. Histone modification results in the majority of histone mediated modulation, which is often alteration of histone’s unprotected amino termini projecting from nucleosome core. Sumoylation, ubiquitination, phosphorylation, methylation as well as acetylation, are the most common histone modifications, with diverse combinations of alterations occurring inside a single nucleosome. H3K4me3 represents the H3 (histone) trimethylation, particularly lysine at position 4, is a mark associated with chromatin which is transcriptionally active, whereas H3K27me3 generates compact chromatin, inhibiting gene expression. The terminology “histone code” refers to elucidating how transcriptional levels are impacted by various histone modification combinations. Identifying proteins that can delete, write or read these marks is critical for understanding the complexity of epigenetic control [21].

#### 2.2.1. Histone Acetylation and Deacetylation

Histone protein’s positive charge can be neutralized by acetylation of lysine residue, via reducing electrostatic association with DNA (which is negatively charged). Reduced association increases the availability of DNA to the protein complexes, resulting in enhanced gene expression. Euchromatin structure is maintained and promoted by acetylation of lysine via complexes of nucleosome-remodeling such as, Swi2/Snf2, with their bromodomains. Although, variables that regulate gene expression are complicated, as well as histone acetylation can also result in decreased gene expression via indirect pathways [66]. Histone deacetylases (HDACs) as well as histone acetyl transferases (HATs) are the two enzyme families responsible for the regulation of lysine acetylation process which occurs at N-terminal tails of core histones. Acetyl CoA is used as a coenzyme by HATs in the process of transferring an acetyl group to epsilon amino group of the lysine side chain. MYST, p300/CBP as well as GNAT are the three families of HATs enzymes. HDACs cause reversal of histone acetylation as well as elevate gene silencing. HDACs exist at DNA methylation site via methyl DNA binding proteins and are the elements of large protein complexes [67]. Tumor development and progression as well as diseases like cardiovascular and neurodegenerative disorders are contributed by dysregulation of HDACs and HATs. There is a sufficient spectrum to make these enzymes appealing therapeutic drug targets [68].

#### 2.2.2. Histone Methylation

Methylation of histone can be mono or demethylated and takes place at lysine residues. Protein arginine methyltransferases (PRMTs) as well as protein lysine methyltransferases (PKMTs) catalyze histone methylation. Although methylation process can be reversed by the protein demethylases. Studies have discovered more than ten PRMTs, greater than fifty PKMTs as well as greater than thirty demethylating enzymes, indicating that methylation of protein is complicated and driving process [69]. Histone methylation has varying effects on the number of methyl groups as well as transcriptional activity, resulting in changed position of amino acid. H3K9me3 as well as H3K9me2 are suppressive while, H3K9me1 mark is activating, in general. H4K20me1, H3K36me2, H3K27me3 as well as H3K9me3 are often associated with transcriptionally suppressed heterochromatin, whereas H3K36me3 and H3K4me3, are associated with active chromatin [69].

### 2.3. Non-Coding RNAs

Noncoding RNAs expression, like as: large RNAs, short RNAs as well as microRNAs, contribute to the epigenetic gene regulation. Noncoding RNAs have the ability to influence both histone modification as well as cytosine methylation in order to silence DNA repeats present in genome. During interaction of nucleotide RNAs with PIWI protein (piwi-interacting RNAs; piRNAs) specific to spermatogenesis, a class of twenty-nine nucleotide RNAs was discovered, that mapped to repetitive DNA sequences and are crucial for silencing long terminal repeat (LTR) retrotransposons, long interspersed elements (LINEs) as well as silencing short-interspersed elements (SINEs) [70,71,72]. Noncoding RNAs have also been implicated in inactivation of X chromosome. Noncoding RNAs of various kinds may have a role in pathogenesis of several malignant tumors, such as oral squamous cell carcinoma, colon tumor and pancreatic tumor. They also affect the physiological processes, for example: morphological development of tooth [3,70]. MicroRNAs are referred to as short noncoding RNA molecules linked to the aetiology of a number of disorders such as: SGC and also affect post-transcriptional gene expression. Noncoding RNAs may function as both oncogenes as well as TSGs [3].

## 3. Epigenetic Alterations in Salivary Gland Tumors

In scope of head and neck neoplasms, Salivary gland carcinomas (SGCs) are particularly prevalent, considering their uncommonness in comparison to other malignancies [73]. These lesions possess assorted or diversified collection of benign as well as malignant tumors exhibiting great variation in their clinical demonstration as well as microscopic presentation, and their biological activity varies in accordance with lesion [74]. On account of their unexpected biological as well as clinical activity, these tumors offer great difficulty in management, which can result in therapeutic failure [44]. SGCs account for 5% of all the head and neck malignancies, including a worldwide yearly incidence ranging from 0.4–13.5 cases per 100 individuals [75]. More than thirty tumors have been identified by world health organization (WHO) stratification of head and neck cancers (HNCs). The mucoepidermoid carcinoma (MEC) have been considered as the most prevalent malignant carcinomas. Multiple investigations on this issue have been conducted; nevertheless, causative elements are still unclear, while chemotherapy, chromosomal rearrangement as well as secondary radiation, can be linked to progression of SGCs.

The epigenetic changes have been proposed as causative variables, however, there have been an infrequent investigation examining their relevance in SGCs [3]. Variations in DNA sequence altering the protein expression, but do not change the order of nucleotide bases are characterized as epigenetic. Epigenetic alterations are important in physiological activities such as replication, transcription as well as DNA repair. As a result, changes in these pathways might result in formation as well as development of several neoplasms [14]. Epigenetic variations have been classified into three types such as: methylation of DNA (Table 1 and Table 2), histone’s structural modifications as well as variable expression of noncoding RNAs such as microRNAs (miRNAs) (Table 1) [14,15,16]. These epigenetic modifications cause a widespread downregulation of gene expression patterns, which leads to progression as well as development of a variety of cancers such as SGCs [15,22,23,24]. Although, these processes are changeable hence, a complete knowledge of these alterations may lead to the identification of novel therapeutic targets for a variety of disorders, such as cancer [14,15,16]. Recent investigations have suggested that these three epigenetic changes might be implicated in formation as well as progression of SGCs [3]. However, the majority of studies have focused on malignant SGCs. It has been established that epigenetic changes can also contribute to benign SGCs [23]. Various approaches as well as models such as (human biopsy tissue and cancer stem cells) are being used to explore these modifications, with genomic research revealing the epigenetic landscape of SGCs [15,23,24,28]. However, several investigations have been conducted, elucidating comprehensive and detailed function of epigenetic modifications in SGCs. A limited review is available that summarizes recent information on this genetic pathway.

### 3.1. DNA Methylation in Mucoepidermoid Carcinomas (MECs)

DNA methylation, a crucial pathway of regulating gene expression, has been linked to neoplasm formation as well as metastasis. In fact, hypermethylation has been proposed as one of the primary pathways for inactivation of tumor suppressor genes (TSGs) [76]. A study conducted by Nikolic et al. in 2018 [23], reported methylation of p53 (TSG) in thirty-five cancer samples as well as repressing of p14ARF (an important regulator of p53 (TSG) function). These events can contribute to the major processes implicated in pathogenesis of mucoepidermoid carcinoma (MECs). However, this is only a qualitative methylation study, and hundred percent of cases indicated epimutations. Even with normal (wild type-unmethylated) p53, a cell would be unable to repair damage if p14 is hypermethylated or otherwise inactivated. Study conducted on aetiology of MECs, by Nikolic et al. [23], confirms significance of epigenetic p14 inactivation, is in line with earlier investigations on carcinoma ex-pleomorphic adenoma (CA-Ex-PA) [77]. Nishimine et al. examined seven MEC specimens for detection of p14 variations and reported no methylation in any specimen, with one deletion. p14 promoter methylation was found to be 19.4 percent while considering intact salivary gland carcinoma (SGC) specimen with dominant adenoid cystic cell carcinoma (AdCC) [78]. Hypermethylation of p14 promotor had been discovered by Ishida and colleagues, in twenty percent specimens of oral squamous cell carcinomas (SCCs). This observation showed a substantial correlation with subsequent clinical presentations, indicating that it might be a critical molecular process in cancer development [79].

Cyclin dependent kinase inhibitors (CKIs) are encoded by p16INK4a TSG, that plays a crucial role in controlling cell-cycle at G1/S phase inspection-point. Lack of a functioning p16 protein result in abnormal regulation of cell-cycle, promoting cancer cell growth. Nikolic et al. [77] observed 60% MEC specimens with hypermethylation of p16, this investigation is consistent with prior studies revealing significance of epigenetic process in SGCs, with rates ranging from 29% to 47% [31,42,59]. Guo et al. [42] investigated thirty-four percent p16 methylation and found no p16 methylation in specimens of MECs. Weber et al., on the other hand, determined that aberrant INK4a-ARF/p53 pathway by various processes found to be a highly common occurrence (84%) in HNC squamous cell carcinomas (SCCs) [59]. Changed levels of TP53 methylation are linked to a variety of cancers such as oral malignancies [80]. Nikolic et al. reported TP53 hypermethylation as an infrequent occurrence in SGCs, but it has increased occurrence of p14 methylation. Specifically, rare TP53 hypermethylation prevents p53 preclusion from pathogenic event, however, instead reinforce the concept that silencing of p14 leads to an inactivation of p53. Hypermethylation of TP53 has been reported by various researchers, in malignant tissues [81], although TP53 hypermethylation as a typical occurrence in healthy cells reported by others [80], which is consistent with ramifications of Nikolic et al.

Function of hTERT gene as well as of telomerase is controlled by significant monitoring systems which include its promotor methylation, but this is occasionally contradicted by generic pattern of DNA methylation as pathways for gene repression [82]. According to Renaud et al., enabled transcription as well as hTERT inhibitors binding arrest are the results of methylated CpG island in hTERT promoter [83]. Nikolic et al. [23], reported malignancies with a significantly greater frequency of hypermethylated hTERT than controls, indicating that formation of MECs may involve this molecular process. Methylation of hTERT is linked with reduced histological grading as well as clinical phases, suggesting that it plays a function in early stages of carcinogenesis. However, methylation of p16 is linked with worse survival rate in HNCs [84], ramifications identical to Nikolic et al. [23], reported that p14 and/or p16 had no effect on survival rate in HNSCCs [85], oral as well as oropharyngeal tumors [86].

The methylation patterns of genes implicated in angiogenesis, cell-cycle regulation and/or DNA repair are frequently abnormal in neoplastic cells [76]. Toyota et al. suggested CpG island methylator phenotype (CIMP). All malignancies were divided into two groups, group 1 is methylation of complete genome and group 2 is the infrequent methylation events, according to the investigators. The first group is more susceptible to promoter methylation-mediated transcriptional suppression of many TSGs [87].

Sasahira et al., discovered that dysregulation of RUNX3 due to DNA hypermethylation as well as protein mislocalization, was substantially linked to metastasis of MEC as well as AdCC, cancer development and SGCs. The promoter region of RUNX3 gets methylated which leads to an inactivation of RUNX3, reported in several carcinomas [43]. SGCs reported to have frequent inactivation of RUNX3 as compared to intact salivary gland tissues, as well as dysregulation of RUNX3 is linked to low survival rate in MEC and AdCC [43].

### 3.2. DNA Methylation in Adenoid Cystic Carcinoma (AdCC)

While we are learning more about genetic changes in AdCC, epigenetic environment is still mostly unclear. An effort has been made by Ling et al. to discover significantly methylated genes to better understand the pathogenesis of AdCC [39]. The modulation of oncogene as well as TSG expression by methylation of DNA promoter is critical in tumorigenesis of AdCC and can contribute to several types of human malignancies [88]. Indeed, AdCC methylome has been analyzed and four genes have been verified [89]. Ling et al., adopted a xenograft-based process involving methylation patterns of complete genome as well as pharmacological demethylation, since viable cell lines were unavailable for the research [90]. Ling et al., reported hypomethylation of HCN2 (potential oncogene) promoter in AdCC cases by using an indifferent inspection for methylated gene promoters [39]. Oncogenic HCN2 may contribute to the AdCC pathogenesis, because of hypomethylation of its promoter. Moreover, HCN2 promoter’s hypomethylation is associated with distant metastasis, recurrence as well as local recurrence of AdCC primary cancers. However, k+ and Na+ are usually conducted by HCN2 [91,92] and HCN2 is permeable to Ca2+. It has been proposed that overexpression of HCN2 may engage in pathogenic Ca2+ signaling [93].

Daa et al., investigated methylation patterns of several CKI genes, focusing on p27 expression [26]. A low occurrence of p27 (26.5%) methylation was reported but, p21, p19, p18 as well as p15 showed high occurrence of methylation ranging from 68.8% to 92.3% [23]. These findings complemented as well as broadened the investigations of Li et al. [31] and Maruya et al. [94], reporting that CKI genes are associated with frequent methylation in AdCC. Promoter methylation suppresses gene expression, considered as a general assumption. As a result, findings of Daa et al., suggested that multiple CKI gene expression may be reduced in AdCC [26]. Furthermore, significant prevalence of CKI gene promoter’s methylation indicates that abnormal methylation occurs early in the course of ACC. Daa et al., reported, nuclear p27 expression in all AdCC specimens including healthy salivary gland cells surrounding AdCC cells [26], validating investigations of Takata et a [95]. Accumulation of p27 is widely recognized in nucleus of tumor cells in the G0 phase, identical to healthy salivary gland cells, as well as acts as a cell cycle inhibitor [96]. Investigations of Daa et al., point to p27 downregulation as a potential contribution to AdCC carcinogenesis [26], with methylation as the likely pathway behind this dysregulation, as has been documented in other malignancies [95,96,97,98,99]. There are various probable explanations for dysregulation of p27 in cancer that is linked to DNA methylation level. Another explanation is that the cancer cells differ in their p27 promoter’s methylation level. In mammalian DNA, methylation occurs at cytosine residues, following cell division by DNA methyltransferase, only certain cancer cells have methylated genes at any one moment. Another possibility is that methylation occurs exclusively in one allele of a gene. p27 would be expressed by various cancer cells in this case, and the PCR product produced in MSP would only use M-primers [26]. 

In several forms of HNCs, including oral squamous cell, laryngeal, thyroid as well as nasopharyngeal, methylation of a promoter is a frequent process underpinning the inactivation of a variety TSGs, including MGMT, hMLH1, DAPK, RASSF1A, p16INK4a as well as E-cad [100]. Methylation of promoters of DAPK, RASSF1A and 16INK4a was found to be frequent in AdCC of salivary gland [31]. Zhang et al., reported methylation of E-cad promoter in 57 percent of individuals having AdCC of salivary glands, with greater incidence of any of the five gene promoters studied in same cancer [28]. Findings of Zhang et al. [28], were similar with previous evidence in which methylation of E-cad promoter was found in seventy percent of individuals with AdCC of salivary gland [35]. Zhang et al., reported E-cad promoter methylation was linked to a decrease in expression of E-cad protein, in AdCC patients, indicating methylation of E-cad causes a dysregulation of E-cad protein [28].

### 3.3. DNA Methylation in Carcinoma Ex-Pleomorphic Adenoma (Ca Ex-PA)

DNA methylation is identified to have a role in tumor progression and growth, as well as determining the methylation level of TSGs constitutes a promising method for initial cancer identification [76]. Cancerous cells frequently have abnormal methylation of several genes, including those that control angiogenesis, DNA repair and cell cycle [101]. There is a scarcity of data on the epigenetics of SGCs, especially the methylation levels of p14 as well as p16. Nikolic et al., investigated p14 as well as p16 TSG promoter’s hypermethylation as a frequent occurrence in CXPA based on its significant incidence, despite its lack of diagnostic significance [77]. DNA methylation level is influenced by ethnicity as well as race, which may elucidate greater prevalence of p14 as well as p16 promoter hypermethylation identified in a study conducted by Nikolic et al. [77]., when compared to earlier publications [102,103,104].

Nikolic et al., suggest that an increase in telomere length or telomeric instability, can contribute to the etiology of CXPA. Importantly, the CXPA showed significant telomere length variability, which is commonly associated with (alternative lengthening of telomeres) ALT phenotype. Findings of Nikolic et al., showed that hypermethylation of p14ARF results in extended telomeres (*p* = 0.013), as well as while identical findings have not been discovered in literature, they support the idea that p14 silencing impacts p53-associated cancer suppression [77].

Gene expression changes are primarily conducted by epigenetic as well as genetic methods. Whereas epigenetic variations cause transcriptional changes, genetic variations generally modify the structure or number of specific gene [105]. Methylation of CpG island in promoter area is a typical epigenetic technique for altering gene expression. This regulation is mostly accomplished by the inactivation of TSGs RASSF1A, DAPK, MGMT as well as p16. In several cancers, the silencing of E-cadherin is primarily determined by modified methylation level of CDH1 promoter [105,106,107]. Silencing of CDH1 is linked to advanced tumor stage and an aggressive character [105]. This is the first study to look at the methylation status of the CDH1 promoter’s methylation level in carcinoma Ex-pleomorphic adenoma (CXPA), first time investigated by Xia et al. [29]. Investigations of Xia et al., found a link between methylation of CDH1 promoter and E-cadherin expression [29]. They identified lack of E-cadherin expression in 35.14% (13/37) of patients with CXPA. These investigations are comparable to the findings of Zhang et al. [49] reporting, negative identification rate of 38.33% in sixty AdCC patients. Although, no expression of E-cadherin was identified in 87.26% (18/23) of oral squamous cell carcinoma (SCC) patients as well as in 68.42% (26/38) of eyelid SCC patients. According to Xia et al., Bisulfite-assisted genomic sequencing PCR (BSP) is considered as the gold standard method which not only identifies the DNA methylation at each CpG location separately but also identifies methylation of CDH1 promoter [29]. Xia et al., investigated 67.57% (25/37) methylation rate of CDH1 in CXPA [29]. This percentage is identical to multiple other carcinomas, such as: colorectal cancer (52%) [108] breast cancer (65–95%) [108,109,110,111] as well as primary lung carcinoma (88%) [112]. Selective DNA methylation has been reported by Xia et al., in the first four CpG regions compared to the remaining CpG regions [29]. 

The relationship between E-cadherin expression as well as CDH1 methylation was determined by Xia et al., in cases with CXPA [29]. In clinical cases, CDH1 methylation was shown to be strongly linked with lower expression of E-cadherin (*p* < 0.001). Furthermore, Xia et al., examined methylation level of CDH1, associated CDH1 mRNA as well as protein levels in SM-AP1 (stromal membrane-associated protein 1) and SM-AP4 cell lines. Cells with increased CDH1 methylation levels expressed less E-cadherin, which was consistent with the previous findings. Although, according to investigations of Xia et al., in each case, decreased levels of E-cadherin expression were not related with promoter methylation of CDH1 [29]. One sample from a low-methylation group showed no expression of E-cadherin. Apart from promoter methylation, there are other pathways which result in suppression of CDH1 production including translational as well as post-translational control, particular transcriptional factors, inactivating gene mutations, loss of heterozygosity (LOH) at 16q22.1 and changes in chromatin structure [113,114,115].

Hence, it can be proposed that DNA methylation in patients with CXPA, predominantly but not completely control levels of E-cadherin expression, both in vitro as well as in vivo. Further research into other regulatory pathways controlling CDH1 in CXPA may be conducted. Reduced expressions of E-cadherin have been linked to tumor recurrence, metastasis and invasion in individuals with breast [111], bladder [116], and oral squamous cell carcinomas [117].

p16INK4a reported no specific variations or microdeletions in any exon in forty-two patients of PA, although twenty eight percent patients reported p16INK4a promoter methylation, which associated with lack of mRNA production. Augello reported fourteen percent hypermethylation of p16INK4a TSG promoter. This disparity between research of Augello et al., and the other data is most likely attributable to dietary influence. As previously documented, TP53 variations as well as p16INK4A methylation arise only in epithelial elements of PAs, indicating that these portions of adenoma may possibly progress into tumor [118].

### 3.4. DNA Methylation in Acinic Cell Carcinoma

Other benign as well as malignant neoplasms were investigated for variations in epigenome of SGCs (Table 1). RASSF1 and retinoic acid receptor beta2 (RARβ2) genes were found to be frequently methylated in salivary duct carcinoma as well as acinic cell carcinoma. The nuclear receptor superfamily includes human RARβ2 as a member of this family, which performs an important function in controlling the effects of retinoic acid on cell proliferation as well as differentiation. Lack of RARβ2 expression has been linked to the development of mammary ductal carcinoma. 

The Histone hypoacetylation H3 (lys9) was shown to be notably prominent in malignant SGCs in contrast to benign cancer in intriguing research incorporating 84 samples of SGCs (42 malignant and 42 benign) [44]. Researchers also found that acetylated tumour cells were less likely to multiply [44]. Thus, unlike in other malignancies like breast cancer as well as pancreatic adenocarcinoma, H3 acetylation has a negative effect on proliferation in SGCs [44]. This is likely because various tissues and organs have unique processes that contribute to tumor formation and can function in opposite ways. 

Salivary ACC is an aggressive kind of SGC, and research into histone changes is progressively advancing towards the quest for novel target therapy [45]. Long-term survival is less encouraging because recurrent tumors and metastasis to distant organs such as liver, bones and lungs are common side effects of therapy for this malignancy [45]. Additional research found that multiple genes involved in chromatin remodeling (such as AT-rich interaction domain containing 5B [ARID5B], lysine-specific demethylase 5A [KDM5A], SW/SNF-related, matrix-associated, actin-dependent regulator of chromatin, subfamily A, member 2 [SMARCA2], and chromodomain helicase DNA-binding protein 2 [CHD2]) were aberrant in 50% of AdCC [119,120]. Mutations in chromatin regulator genes were found in 35% of ACC tumors in another investigation [102].

Out of the 22 malignant SGT patients, Pouloudi et al., reported that 14% were HDAC-1 positive (3 cases), 82% were HDAC-2 positive (18 cases), 36% were HDAC-4 positive (8 cases), and 18% were HDAC-6 positive 4 cases [121]. However, Pouloudi et al., did not find any significant relationships between gender or age of the patients and HDAC expression, previous research has revealed similar links across a variety of carcinoma types. Specifically, research has linked male genders and younger age group of patients to elevated HDAC-1 expression in mobile tongue SCC [122], whereas in gastric carcinoma [123] it has been linked to older age group of patients. By contrast, instances of invasive ductal breast cancer where HDAC-6 expression was high had younger individuals [124].

### 3.5. Non-Coding RNAs Contributing to SGCs

Expression of many different genes may be influenced by a class of RNA (transcribed) molecules called noncoding RNAs (ncRNAs) [15,16,125,126]. SGCs may be more aggressive and develop at a faster rate if miRNAs play a role in their development [46]. One of the most upregulated miRNAs in MEC tissues was miR-302a, and in-vitro, its overexpression in SGT cell lines led to cancer cell invasion [46]. The t (11;19) translocation that causes *CRTC1-MAML2* gene fusion is a significant oncogenic driver for the development of MECs that cause various disorders [127]. Recent research has identified a particular lncRNA (LINC00473) as a key regulator of the oncoprotein CRTC1-MAML2 [47]. Furthermore, bioinformatic research found that over 3091 lncRNAs were changed throughout the pathogenesis of MEC; however, the clinicopathologic relevance of these results required additional study [52]. Additionally, miR-34a and miR-21 were elevated, while miR-20a was downregulated, and MEC lacked the majority of related-apoptosis transcripts, when studying the expression of transcripts and miRNAs implicated in the regulation of apoptosis in MECs [51]. Furthermore, interaction of other genes involved in this malignancy with lncRNAs has also been reported, however the mechanism associated with this connection is not entirely known [50].

### 3.6. Tumour to Tumour Interactions

Tumour to tumour interaction plays critical role in the carcinogenesis. Imperatively, there are multiple factors and proteins produced by the tumour and these tumoural markers create a complex interaction in the tumour ecosystem. The interaction of some of these markers in the tumour microenvironment has significant impact on the tumour biology and characteristics. For instance, the intratumoural lymphocytes has been associated with high risk of neck nodes metastasis and high-grade tumours in acinic cell carcinoma [128]. This is likely due to the presence of PDL1 expression in higher level in association with elevated immune cell infiltration of T and B cells. This underlies why the acinic cell carcinoma has been shown to have unfavourable prognosis and has higher risk of lymph node metastases.

Additionally, the process of cancer metastases is important as it influence the prognosis of patient with SGCs. The cancer cells infiltrate the lymphatic and blood vessels through the migration of extracellular matrix, where the main enzyme systems of MMPS is required, and this is located in the invadopodia of cancer cells. In adenoid cystic carcinoma, Lissencephaly 1 (L1S1) regulates the invadopodia formation and has been shown to associate with matrix metalloproteinases (MMPs) expression. Lissencephaly L1S1 is a microtubule associated protein which regulates the microtubules stability, and it can mitigate the metastatic potential of ACC through the invadopodia formation and ECM degradation [129]. Also, integrin linked kinase (ILK), play important role in ECM interactions, with presence of other cofactors such as growth factors and integrin, which regulates cells differentiation, migration and apoptosis. This has positive roles in tumour progression and transformation [130].

## 4. Epigenetic Biomarkers for Diagnosis, Prognosis and Treatment Response Prediction of SGCs

Pathological and clinical factors, such as therapy response, the possibility of local recurrence, distant metastasis, perineural invasion as well as slow development are used to determine the SGCs prognosis [131,132]. Finding additional indicators to help in evaluating neoplasm prognosis is still necessary owing to the heterogeneous histopathologic characteristics of this carcinoma. Therefore, it is very crucial to understand the contribution of epigenetic variations in SGCs, given that these variations can be employed as prognostic indicators and that techniques have been advanced at molecular level, to the point where they have aided in identifying molecular signatures associated with other neoplasms. Several potential epigenetic biomarkers have been reported for diagnosis, prognosis and treatment response prediction of SGC have been (Table 3).

In AdCC, one significant gene exhibited some prospects as a potential biomarker, linked with histologic cancer grade and patient survival, named as engrailed homebox 1 gene (EN1) [133]. In AdCC, Aquaporin-1 (AQP1) and Suprabasin (SBSN), may be intriguing entrants for molecular identification, when epigenetic oncogenes were screened genome-wide [38]. In AdCC, SBSN is crucial for preserving epidermal differentiation as well as cell proliferation, which is anchorage-dependent and independent, whereas water transportation across the membrane is accompanied by AQP1 [38]. In AdCC, both are hypomethylated [37,38]. Patients who had hypomethylated SBSN had a higher chance of regional recurrence, but individuals having hypomethylated AQP1 had enhanced overall survival [37,38]. As a result, both genes may represent possible molecular indicators for SGT prognosis.

The prognosis of SGCs has been affected by the inactivation of E-cadherin which leads to metastasis as well as aggressive carcinoma [28]. It has been reported that promoter methylation causes the inactivation of E-cadherin in AdCC of head and neck [28]. Promoter methylation causes this protein might be playing a significant role in perineural invasion and cancer cell differentiation, and these characteristics may be crucial in determining the prognosis of various malignancies [28]. Methylated Ras association domain family protein1 isoform A (RASSF1A) is a potential indicator of poor prognosis and survival associated with this neoplasm as well as involved in the initiation, differentiation and advancement of AdCC, as reported by the findings of Chinese population research [134].

The AdCC has been the most often researched SGT due to its propensity for perineural invasion, metastasis and aggressiveness. There is a need to find prognostic indicators in order to enhance the overall survival as well as treatment of patients harboring AdCC. Despite the fact that methylation has been the utmost extensively researched epigenetic mechanism, it has been reported that distant metastasis, as well as fast cell proliferation, are a significant predictor of AdCC caused by higher methylation levels of H3K9me3 [45]. Additionally, this histone methylation may serve as a helpful prognostic marker for the head and neck AdCC prognosis, when linked with histologic characteristics [45]. Despite the fact that these results are encouraging, they still need correlation to clinical data and more research to be confirmed as molecular prognostic indicators in SGCs.

## 5. Epigenetic Drugs in SGC

Surgery and radiation, whether or not in conjunction with chemotherapy, are the mainstays of treatment for SGCs [14,73]. The poor overall survival rates shown over lengthy periods of time, however, indicate that more work needs to be done in this area [44]. To develop more effective therapeutic approaches, it is crucial to increase our understanding of clinical implications, etiopathogenesis, and biological behavior of these pathways in SGCs [44]. Drugs that disrupt epigenetic processes implicated in the onset and development of certain malignant neoplasms have become more prevalent during the last ten years (Table 4) [14]. However, there are still a few studies that use SGCs as a target for epi-drug treatment.

Despite the paucity of research in this area, it was shown that the proliferation of these cancers is affected by H3 acetylation in an inversely proportionate way, and medications that inhibit these processes may be useful in these tumors [44]. The histone acetyltransferase inhibitors (HDACi) administration was encouraging in enhancing cisplatin-based treatment, as well as disrupting cancer stem cells (CSCs), and the involvement of CSCs in radioresistance, chemoresistance phenotype, and cancer development has been reported by in-vitro research, in MECs [44]. For the treatment of a number of cancer types, including solid tumors, glioblastoma multiforme, and haematological malignancies, there is a need to study the drug development that changes the epigenetic processes implicated in histone confirmation [14].

In experimental investigations for SGC treatment, target treatments for DNMTs are being developed [135]. Various DNMT-targeting medications, such as 5-aza-2′deoxycytidine (5-aza-dC), have previously been created. The fundamental role of the target medications is limiting the activity of DNA methyltransferases since there is a rise in these enzymes’ activities in malignancies. Using AdCC cell lines, it was previously shown that 5-aza-dC suppressed cancer cell invasion by reversing hypermethylation of RECK (TSG). This suggests that 5-aza-dC may be a potential chemotherapeutic strategy for AdCC treatment [135].

## 6. Conclusions

The epigenetic processes have significant roles in the development as well as progression of SGCs. Regardless of the fact that additional epigenetic modifications are known already, the most investigated mechanism in SGCs is DNA methylation, followed by histone conformational abnormalities as well as non-coding RNAs. The TSG (p14) hypermethylation proves to be the most important event in the formation of MECs and cell cycle disruption caused by epigenetic abnormalities is found to be the most significant event in the development of AdCC. These significant molecular markers are paramount as it could be used as pivotal diagnostic biomarkers for each of the types of malignant SGCs and it can also be used for patient prognostic stratification. Subsequently, a more refined and focus therapeutic approaches can be implemented for each of these SGCs.

Additionally, the CKI genes hypermethylation has also been reported to be a frequent process in the progression of AdCC, whereas HCN2 hypomethylation has been reported to be the possible biomarker that point out the more aggressive form of AdCC. All of these molecular alterations might be exploited as sophisticated prognostic molecular markers and could assist in fine tuning the management of SGCs. This is critical as malignant SGCs have different prognosis and requires specific treatment approach. At near future, with the wide availability of these versatile molecular biomarkers, the treatment of SGC patients can be escalated.

## Figures and Tables

**Figure 1 cancers-15-02111-f001:**
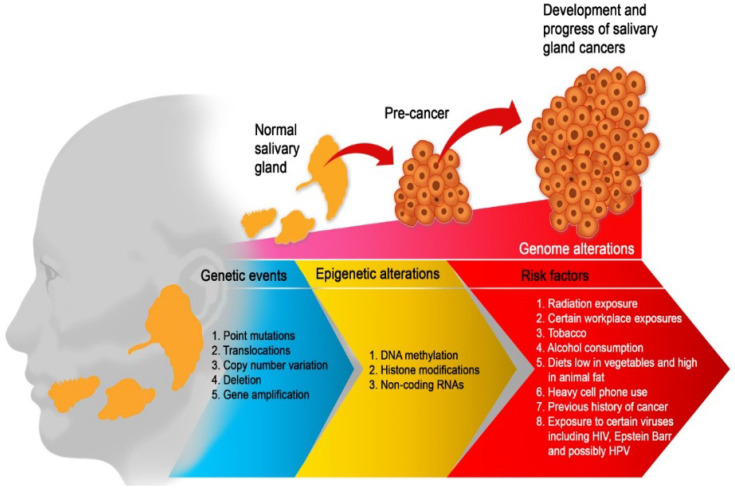
Etiological factors of salivary gland carcinoma. Multiple factors are working together to drive SGC from a few aberrant cells to a tumour phenotype with the capacity to metastasis. Therefore, the optimum environment for malignant development is maintained by a complex interplay of genetic events, risk factors, and epigenetic mechanisms. All of these factors work together to promote an unstable genome and hence, promote cancer progression.

**Figure 2 cancers-15-02111-f002:**
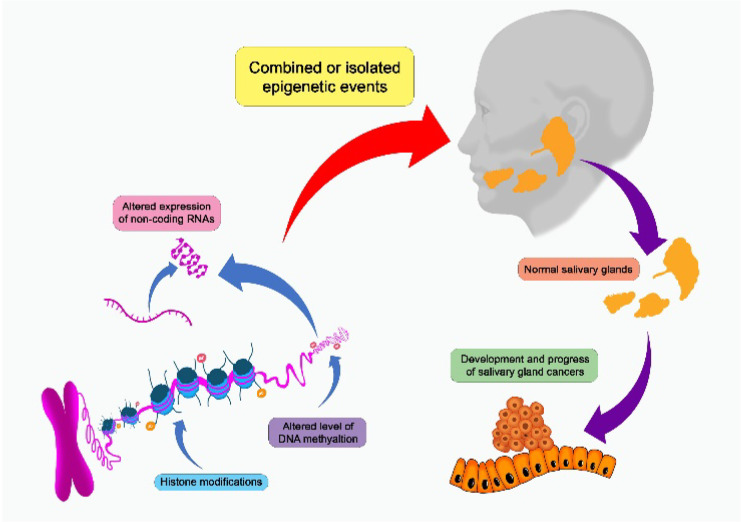
Salivary gland carcinomas can be affected by a number of epigenetic events that can alter the development and progression of the cancer.

**Table 1 cancers-15-02111-t001:** Epigenetic markers associated with SGCs.

First Author/Year	Country	Gene/Genome Elements	Genome Region	SGC Tumor (Malignant and/or Benign)	Sample Type	Molecular Alteration	Biological Function Associated with Molecular Alteration	References
DNA Methylation								
Nikolic et al., 2018	Serbia	p14^ARF^/p16^INK4a^	9p21.3	Pleomorphic Adenoma/Carcinoma Ex Pleomorphic Adenoma	formalin-fixed, paraffin-embedded (FFPE) samples	hypermethylation	Transcriptional silencing of the *p14/ARF* gene	[23]
Wang et al., 2015	USA	CLIC3	9q34.3	Mucoepidermoid carcinomas (MECs)	FFPE MEC tumor samples	hypermethylation	Oncogenic ole	[24]
Shieh et al., 2005	Taiwan	CDH1	16q22.1	Mucoepidermoid carcinomas (MECs)	FFPE MEC tumor samples	hypermethylation	Loss of E-cadherin expression	[25]
Daa et al., 2008	Japan	p15, p18, p19, p21, & p27	9p21.3, 1p32.3, 19p13.2, 6p21.2, 12p13.1	Adenoid cystic carcinoma (AdCC)	FFPE AdCC tumor samples	hypermethylation	Cell cycle disruption	[26]
Williams et al., 2006	Texas	DAPK, MGMT, RARβ2, anRASSF1 (Ras association domain family protein1 isoform A)	9q21.33,	Adenoid Cystic Carcinoma, Mucoepidermoid Carcinoma, Acinic cell Carcinoma, Salivary Duct Carcinomas	FFPE tissues from Adenoid cystic carcinoma (AdCC), Mucoepidermoid carcinoma (MECs), Salivary duct carcinoma (SDCs), and acinic cell carcinoma samples	*RASSF1* (Ras association domain family protein1 isoform A) and *RARβ2* were highly methylated in malignant tumors and *MGMT* and *DAPK* in benign neoplasm.	Low or absent protein expression	[27]
Zhang et al., 2007	China	E-cadherin	Cadherin 1: 16q22.1	Adenoid cystic carcinoma (AdCC)	Tissue samples	Hypermethylation	E-cadherin plays a critical role in transducing signals to influence several important biologic processes. Reduced expression of E-cad, caused by genetic and epigenetic events, has been observed in aggressive carcinoma types. promoter methylation of E-cadherin is a more common mechanism for its inactivation.	[28]
Xia et al., 2017	China	CDH1	CDH1 gene located on 16q 22.1	Salivary carcinoma ex pleomorphic adenoma (CXPA)	Formalin-fixed and paraffin-embedded tissues	Promoter hypermethylation	CDH1 silencing is directly related to advanced tumor stage and an aggressive phenotype. The association of CDH1 methylation with cervical lymph node metastasis, histological grade and advanced tumor stage suggests that the CDH1 gene may be particularly important in salivary CXPA tumor progression.	[29]
Lee et al., 2008	Republic of Korea	RARβ2 and RASSF1	chromosomal region 3p24 and chromosome 3p21.3	AdCC, adenoid cystic carcinoma, and salivary duct carcinoma.	Tissue samples	Promoter hypermethylation	These genes are tumor suppressor genes and known for their ability to suppress vital cellular processes, including cell-cycle regulation, apoptosis, DNA repair, differentiation, and metastasis. The hypermethylation of the two genes may synergistically involve in the carcinogenesis of these two entities.	[30]
Li et al., 2005	Texas	p16^INK4a^, RASSF1A, and DAPK	chromosome 9p21, chromosome 3p21. 3 and chromosome 9q21.33	Adenoid cystic carcinoma (AdCC)	Formalin-fixed and paraffin-embedded tissues	Promoter hypermethylation	Promoter methylation of these gene often results in silencing of its expression and is acommon mechanism to inactivate tumor suppressor genes in tumorigenesis.	[31]
Durr et al., 2010	United States of America	APC, Mint 1, PGP9.5, RAR-b, andTimp3	Chromosome 5q22. 2, SPEN gene, chromosome 4p14, chromosome 17q21.2 and Chromosome 22	Malignant SGTs	Paraffin embedded tissues	Promoter hypermethylation	Promoter methylation of these gene may contribute to salivary gland carcinogenesis	[32]
Uchida et al., 2004	Japan	14-3-3 σ	chromosome 8q22.3	Adenoid cystic carcinoma (AdCC)	Tissue sample	Hypermethylation	The downregulation of 14-3-3 σ by hypermethylation of the CpG island may contribute to salivary gland carcinogenesis	[33]
Fan et al., 2010	China	PTEN	Chromosome 10q23. 31	Adenoid cystic carcinoma (AdCC)	ACC-2 cell lines	Promoter hypermethylation	The hypermethylation of the PTEN promoter region is one of the major mechanisms leading to reduced expression of PTen in adenoid cystic carcinomas. This indicates that PTen is an important candidate gene involved in the pathogenesis of adenoid cystic carcinomas	[34]
Maruya et al., 2004	Japan	E-cadherin	Cadherin 1: 16q22.1	Adenoid cystic carcinoma (AdCC)	Paraffin-embedded tumor tissues	Hypermethylation	E-cadherin plays a critical role in transducing signals to influence several important biologic processes. Reduced expression of E-cad, caused by genetic and epigenetic events, has been observed in high grade and aggressive tumors. promoter methylation of E-cadherin is a more common mechanism for its inactivation.	[35]
Kanazawa et al., 2018	USA	GALR1 and GALR2	G-protein coupled receptors family	Salivary duct carcinoma (SDC)	Tissue samples	Promoter hypermethylation	The galanin receptors, GALR1 and GALR2, are members of the GPCR superfamily, and serve as important tumor suppressor genes. The silencing of the GALR1 and GALR2 genes by methylation may constitute a critical event in SDC.	[36]
Tan et al., 2014; Shao et al., 2011	USA	AQP1	Aquaporins: located on chromosome 6 in a region with homology of synteny with human 7p14.	Adenoid cystic carcinoma (AdCC)	Tumor tissue sample	Hypomethylation	AQP1 is a small transmembrane protein that selectively transports water across cell membranes. It is highly expressed in several tumor types and has been implicated in tumor cell proliferation, extravasation, migration, and metastasis. AQP1 was significantly hypomethylated in ACC tumors compared with controls	[37,38]
Ling et al., 2016	USA	HCN2	Chromosome 19	Adenoid cystic carcinoma (AdCC)	Formalin-fixed, paraffin-embedded tissue sample	Promoter hypomethylation	HCN2 normally conducts K+ and Na+, it has been reported that HCN2 was permeable to Ca2+, and it was suggested that they may participate in pathological Ca2+ signaling when HCN2 is overexpressed. Hypomethylation of HCN2 as a potential biomarker for ACC that may be associated with more aggressive disease	[39]
Ge et al., 2011	China	RUNX3 gene	Chromosomal region 1p36	Adenoid cystic carcinoma (AdCC)	Tissue samples	Hypermethylation	The methylation of the promoter 5′-CpG island in the RUNX3 gene is a major gene-silencing mechanism. RUNX3 protein expression is significantly related to metastasis and T stage.	[40]
HU et al., 2011	China	P16	Chromosome 9p21	Carcinoma ex Pleomorphic adenoma (Ca-ex-PA)	Tissue samples	Promoter hypermethylation	Overexpression of p16 protein in the cytoplasm and decreased expression of p16 protein in the nucleus may play important roles in the evolution of pleomorphic adenoma to Ca-ex-PA	[41]
Guo et al., 2007	China	p16INK4a	Chromosome 9p21	Mucoepidermoid carcinoma (MEC)	Tissue samples	promoter hypermethylation	Alterations of the p16INK4a tumour suppressor gene are often observed in a variety of human cancers and are considered to play a critical role in the transition to malignant growth. the main mechanisms of inactivation of the p16INK4a gene in MEC of the salivary glands are promoter hypermethylation.	[42]
Sasahira et al., 2011	Japan	RUNX3	Chromosomal region 1p36	Pleomorphic adenoma (PA). Adenoid cystic carcinoma (AdCC) andMucoepidermoid carcinoma (MEC)	Formalin-fixed, paraffin embedded samples	Hypermethylation	RUNX3 inactivation is observed more frequently in salivary gland tumors than in normal salivary gland tissues and RUNX3 downregulation is significantly correlated with tumor progression and poor prognosis in AdCC and MEC	[43]
Histone modifications								
Wanger et al., 2017	Brazil	H3lys9	histone H3 at Lys 9	Adenoid cystic carcinoma, mucoepidermoid carcinomas and acinic cell carcinoma	FFPE tissue blocks of SGTs	Histones are hypoacetylated	histone deposition, chromatin assembly and gene activation	[44]
Xia et al., 2013	China	H3K9me3 and H3K9Ac	Trimethylation of histone 3 lysine 9 1q42.13	Adenoid cystic carcinoma (AdCC)	AdCC tumor samples	Histones were acetylated and methylated	Rapid cell proliferation and distant metastasis in ACC	[45]
Non-coding RNAs								
Binmadi e al., 2018	Maryland	miR-302a	4q25	Mucoepidermoid carcinomas (MECs)	MECs tumor samples	Non-coding RNA was upregulated	Cancer invasions	[46]
Brown et al., 2019	Finland	miRNA-150, miRNA-375, and miRNA-455-3p	Chromosome 19q13, chromosome 2 and chromosome 9 at locus 9q32	Mucoepidermoid carcinomas (MEC) and Adenoid cystic carcinoma (AdCC)	Formalin-fixed paraffin embedded	miRNA-150 and miRNA-375 expression was significantly decreased in AdCC and PAC, whilst miRNA-455-3p showed significantly increased expression in AdCC when compared to PAC.	The post-transcriptional protein expression has been shown to play important roles in neoplastic and non-neoplastic processes. These non-coding RNAs presented alternated expression in SGC.	[47]
Mitani et al., 2013	USA	miR-17 and miR-20a	Chromosome 13	Adenoid cystic carcinoma (AdCC)	Tissue samples	Overexpression of the miR-17 and miR-20a	miRNAs play a role in the regulation of cellular pathways in the ACC tumorigenesis, and this may be influenced by the fusion gene status. Overexpression of the miR-17 and miR-20a were significantly associated with poor outcome in the screening and validation sets	[48]
Persson et al., 2009	The Netherlands	miR-15a/16 and miR-150	Chromosome 13q14 and chromosome 19	Adenoid cystic carcinoma (AdCC)	Tissue samples	miR-15a/16 was overexpressed in ACC as compared with normal glandular tissues, whereas the expression of miR-150 was lower in ACC than in normal glandular tissues	The deregulation of the expression of MYB and its target genes is a key oncogenic event in the pathogenesis of ACC. miR-15a/16 and miR-150 recently were shown to regulate MYB expression negatively. The MYB-NFIB fusion is a hallmark of ACC and that deregulation of the expression of MYB and its target genes is a key oncogenic event in the pathogenesis of ACC	[49]
Xu et al., 2019	China	Different lncRNA and mRNA in PLAG1 gene	Chromosome 8q12	Carcinoma Ex Pleomorphic Adenoma	Mouse tumors glands	lncRNAs and mRNAs were differentially expressed in PA tissues obtained from PLAG1 transgenic mice as compared with those from control mice.	The differentially expressed mRNAs and lncRNA revealed that these mRNAs were closely associated with a number of processes involved in the development of PA.	[50]
Flores et al., 2017	Brazil	miR-15a, miR16, miR-17-5p, miR-21, miR-29, miR-34a and miR-20a	Chromosome 13q14, Chromosome 13q14, chromosome 13, chromosome 17q23.2, chromosome 7q32.3, chromosome 1p36.22 and chromosome 19	Mucoepidermoid carcinoma (MEC) and Carcinoma Ex Pleomorphic adenoma (PA)	Tissue samples	The expression of miR-21 and miR-34a was upregulated in MEC, respectively. Downregulation of miR-20a was observed in PA and in MEC. The upregulation of miR-15a, miR16, miR-17-5p, miR-21, miR-29, and miR-34a was observed in PA	The expression of apoptosis-regulating miRNAs in salivary gland tumors, suggesting possible involvement of these microRNAs in salivary gland tumorigenesis.	[51]
Lu et al., 2019	China	hsa_circ_00123, NON- HSAT154433.1 and circ012342	circRNA	Mucoepidermoid carcinoma (MEC)	Tissue samples	The circR- NAs showed the highest fold change in MEC group compared with normal control group. The elevated expression of NON- HSAT154433.1 and decreased expression of circ012342 were observed and closely related to the pathogenesis of MEC	An increasing number of circRNAs have been discovered in various diseases and exhibit cell-type or tissue-specific patterns.	[52]
Lam-Ubol et al., 2022	Thailand	H3K9Me3 and H3K9Ac	Trimethylation of histone 3 lysine 9 1q42.13	Mucoepidermoid carcinoma (MEC) and Adenoid cystic carcinoma (AdCC)	Tissue samples	Hyperacetylation and trimethylation of histone H3	Increased H3 trimethylation at lysine residue 9, as well as H3 acetylation at lysine residue 9 and 18, could be involved in the progression of malignancies.	[53]

Abbreviations: Mucoepidermoid carcinoma (MEC), Adenoid cystic Carcinoma (AdCC), Salivary duct carcinoma (SDC), Pleomorphic adenoma (PA), Carcinoma Ex-Pleomorphic adenoma (CA Ex-PA).

**Table 2 cancers-15-02111-t002:** Description of methylated genes contributing to the pathogenesis of SGCs.

Carcinoma Type	Methylated Genes	References
Mucoepidermoid Carcinoma	P14, CLIC3, CDH1, APC, Mint1, PGP9.5, Timp3, p16(INK4A), RUNX3, DAPK, MGMT, RARβ2 and RASSF1	[3,11,23]
Adenoid cystic carcinoma	P15, p18, p19, p21, APC, Mint1, PGP9.5, Timp3 Cyclin-dependent kinase inhibitors (p27), HCN2, AQP1, SBSN, RUNX3, DAPK, MGMT, RARβ2 and RASSF1	[3,26,28,31]
CA-Ex-PA	RASSF1, p53, p16(INK4A), promoter methylation in CDH1, P14ARF	[29,32,41]
Acinic cell carcinoma	RASSF1 (Ras association domain family protein1 isoform A) and RARβ2, DAPK, and MGMT	[3]

**Table 3 cancers-15-02111-t003:** Diagnostic and prognostic biomarkers in SGCs.

Potential Diagnostic Biomarkers					
Gene/Genome Elements	Genome Region	Sample Type	Molecular Alteration	Tumor Types of SGC (Malignant and/or Benign)	References
RARβ2 and RASSF1A	Chromosome 3p21. 3 and Chromosome 3p24	Fresh-frozen tissue specimens	Hypermethylation	Salivary Gland Carcinomas (AdCC, adenoid cystic carcinoma, and salivary duct carcinoma)	[30]
p16^INK4a^	Chromosome 9p21	Tissue samples	Promoter hypermethylation	Mucoepidermoid carcinoma (MEC) and adenoid cystic carcinoma (AdCC)	[31,42]
p15, p18, p19, p21, and p27	Chromosome 9p21.3, 1p32.3, 19p13.2, 6p21.2, 12p13.1	Tissue samples	Promoter hypermethylation	Adenoid cystic carcinoma (AdCC)	[26,94]
APC, Mint 1, PGP9.5, RAR-b, andTimp3	Chromosome 5q22. 2, SPEN gene, chromosome 4p14, chromosome 17q21.2 and Chromosome 22	Tissue samples	Hypermethylation	Malignant SGTs (Pleomorphic adenoma (PA), Mucoepidermoid carcinoma (MEC), adenoid cystic carcinoma AdCC and Salivary duct carcinoma (SDC)	[32]
14-3-3 σ	chromosome 8q22.3	Tissue samples	Hypermethylation	Adenoid cystic carcinoma (AdCC)	[33]
AQP1	Aquaporins: located on chromosome 6 in a region with homology of synteny with human 7p14.	Tumor tissues	hypomethylation	Adenoid cystic carcinoma (AdCC)	[38]
SBSN	Suprabasin: 19q13.12	The saliva of AdCC patients	hypomethylation	Adenoid cystic carcinoma (AdCC)	[38]
acetyl-H3 (lys9)	histone 3 (H3) acetylation at Lys9	paraffin-embedded tissue	hypoacetylated	Adenoid cystic carcinoma (AdCC), Mucoepidermoid carcinoma (MEC) and Adenoid cystic carcinoma (AdCC)	[44]
PTEN	Chromosome 10q23. 31	ACC-2 cell lines	Promoter hypermethylation	Adenoid cystic carcinoma (AdCC)	[34]
RASSF1 and RARβ2	Chromosome 3p24 and chromosome 3p21. 3	Tissue samples	hypermethylation	Adenoid cystic carcinoma (AdCC) and Salivary duct carcinoma (SDC)	[27]
E-cadherin	Chromosome 16q22. 1	Tissue samples	Promoter hypermethylation	Adenoid cystic carcinoma (AdCC)	[35]
p16^INK4^a, RASSF1A, and DAPK	chromosome 9p21, chromosome 3p21. 3 and chromosome 9q21.33	Formalin-fixed and paraffin-embedded tissues	Promoter hypermethylation	Adenoid cystic carcinoma (AdCC)	[31]
P14, p16, hTERT and TP53	chromosome 9p21, chromosome 9p21, chromosome 5p15. 33	formalin-fixed, paraffin-embedded sample	Hypermethylation	Mucoepidermoid carcinoma (MEC)	[23]
CLIC3	Chromosome 9	Tissue samples	Promoter hypomethylation	Mucoepidermoid carcinoma (MEC)	[24]
HCN2	Chromosome 19	Formalin-fixed, paraffin-embedded tissue sample	Promoter hypomethylation	Adenoid cystic carcinoma (AdCC)	[39]
RUNX3 gene	Chromosomal region 1p36	Tissue samples	Hypermethylation	Adenoid cystic carcinoma (AdCC)	[40]
P16	Chromosome 9p21	Tissue samples	Promoter Hypermethylation	Carcinoma ex Pleomorphic adenoma(Ca-ex-PA)	[41]
MiR-455-3p	Chromosome 9 at locus 9q32	Formalin-fixed paraffin embedded	Significantly increased expression in AdCC	Adenoid cystic carcinoma (AdCC)	[47]
Different lncRNA and mRNA in PLAG1 gene	Chromosome 8q12	Mouse tumors glands	lncRNAs and mRNAs were differentially expressed in PA tissues obtained from PLAG1 transgenic mice as compared with those from control mice.	Pleomorphic adenoma (PA)	[50]
**Potential biomarkers of treatment response and prognostic**					
**Gene/Genome elements**	**Genome region**	**Sample type**	**Molecular alteration**	**Tumor types of SGC (malignant and/or benign)**	**References**
EN1 gene	Engrailed Homeobox 1: 2q14.2	The saliva of AdCC patients	hypermethylation	Adenoid cystic carcinoma (AdCC)	[133]
SBSN	Suprabasin: 19q13.12	paraffin-embedded samples	hypomethylation	Adenoid cystic carcinoma (AdCC)	[38]
AQP1	Aquaporins: located on chromosome 6 in a region with homology of synteny with human 7p14.	Tumor tissues	hypomethylation	Adenoid cystic carcinoma (AdCC)	[38]
inactivation of E-cadherin, encoded by CDH1	Cadherin 1: 16q22.1	Tissue samples	Promoter hypermethylation	Adenoid cystic carcinoma (AdCC)	[28]
RASSF1A	Ras Association Domain Family Member 1: 3p21.31	The saliva of AdCC patients	hypermethylation	Adenoid cystic carcinoma (AdCC)	[134]
H3k9me3	9^th^ lysine residue of the histone H3 protein and is often associated with heterochromatin.	The saliva of AdCC patients	trimethylation of histone 3 lysine 9	Adenoid cystic carcinoma (AdCC)	[45]
galanin receptors (GALRs); GALR1 and GALR2	G-protein coupled receptors family	Tumor specimens	Hypermethylation	Salivary duct carcinoma (SDS)	[36]
RARβ2 and RASSF1A	Chromosome 3p24 and chromosome 3p21. 3	Fresh-frozen tissue specimens	Promoter hypermethylation	Malignant Salivary Gland Carcinomas (ACC, adenoid cystic carcinoma, and salivary duct carcinoma)	[30]
E-cadherin	Cadherin 1: 16q22.1	Tissue samples	Promoter hypermethylation	Adenoid cystic carcinoma (AdCC)	[35]
CDH1	Chromosome 16q22.1	Formalin-fixed and paraffin-embedded tissues	Promoter hyperrmethylation	Salivary carcinoma ex pleomorphic adenoma (CXPA)	[135]
RASSF1A	Chromosome 3p21. 3	Formalin-fixed and paraffin-embedded tissues	Promoter hypermethylation	Adenoid cystic carcinoma (AdCC)	[31]
14-3-3 σ	Chromosome 8q22.3	Tissue samples	Hypermethylation	Adenoid cystic carcinoma (AdCC)	[33]
H3K9Ac, H3K9Me3 and H3K18Ac	Trimethylation of histone 3 lysine 9 1q42.13 and histone H3 lysine 18	Tissue samples	Hyperacetylation and trimethylation of histone H3	mucoepidermoid carcinoma (MEC) and adenoid cystic carcinoma (AdCC)	[136]
RUNX3 gene	Chromosomal region 1p36	Tissue samples	Hypermethylation	Adenoid cystic carcinoma (AdCC) and Mucoepidermoid carcinoma (MEC)	[40,43]
miR-17 and miR-20a	Chromosome 13	Tissue samples	Overexpression of the miR-17 and miR-20a	Adenoid cystic carcinoma (AdCC)	[48]
Different lncRNA and mRNAin PLAG1 gene	Chromosome 8q12	Mouse tumors glands	lncRNAs and mRNAs were differentially expressed in PA tissues obtained from PLAG1 transgenic mice as compared with those from control mice.	Pleomorphic adenoma (PA)	[50]
hsa_circ_00123 and NON-HSAT154433.1	circRNA	Tissue samples	The circR-NAs showed the highest fold change in MEC group compared with normal control group. The elevated expression of NON-HSAT154433.1 and decreased expression of circ012342 were observed and closely related to the pathogenesis of MEC	Mucoepidermoid carcinoma (MEC)	[52]

**Table 4 cancers-15-02111-t004:** Potential epigenetic drugs in SGC.

Agent(s)	Cancer Type(s)	Target	FDA-Approved Date	Trial Details	Trial Identifer/Status	Reference
Azacytidine (DNMT inhibitor)		HPV-positive HNSCC (resectable disease)	pending	2014-ongoing (window study)	Recruiting	[137]
Decitabine (DNMT unhibitor)		HPV-positive Anogenital and HNSCC (R/M)	Pending	2019-ongoing (phase 1b)	Recruiting	[138,139]
Cabozantinib	All histologies	c-MET	17 September 2021	Phase II	Active, not recruiting	[140]
Nivolumab	All histologies	PD-1	4 March 2022	Phase II	Active, not recruiting	[140,141]
ivolumab + ipilimumab	All histologies	PD-1 CTLA-4	26 May 2020	Phase II	Active, not recruiting	[142]
Pembrolizumab	All histologies	PD-1	26 July 2021	Phase II	Recruiting	[142,143]
Nivolumab + ipilimumab	All histologies	PD-1 CTLA-4	26 May 2020	Phase II	Recruiting	[142]
Pembrolizumab + lenvatinib	All histologies	PD-1 VEGFR	10 August 2021	Phase II	Not yet Recruiting	[142]
Lutetium-177 PSMA	All histologies	PSMA	23 March 2022	Phase II	Not yet Recruiting	[142]
Axitinib + Avelumab	Adenoid cystic only	VEGFR PD-L1,	14 March 2019	Phase II	Recruiting	[142]
CB-103	Adenoid cystic carcinoma + other tumors	NOTCH	-	Phase I/II	Recruiting	[142]
BB1503 (amcarsetinib)	All histologies + other tumors	NOTCH	5 April 2022	Phase Ib/II	Active, not recruiting	[142]
AL101	Adenoid cystic only	NOTCH	16 September 2021	Phase II	Recruiting	[142]
TeTMYB + BGBA 17	Solid Tumors	MYB	-	Phase I	Not Yet Recruiting	[142]
Transtuzumab	Salivary duct carcinoma + other solid tumours	HER2 positive	2002	Phase II	Active	[144,145]

## Data Availability

Not applicable.

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
