# Peer review of "The Epigenesis of Salivary Glands Carcinoma: From Field Cancerization to Carcinogenesis"

_cancers, 2023, doi:10.3390/cancers15072111_

Round 1
Reviewer 1 Report
Thanks for submitting this extensive overview. I found it a bit disorganized and confusing.
First, the authors start by providing a lengthy irrelevant introduction from which they promise to investigate the epigenetic alternations in frequently diagnosed lesions. However, the selected lesions do not fit together. The clustering of lesions into benign and malignant is way below the concurrent standards. I recommend that the manuscript should be reshuffled to focus on carcinomas ex pleomorphic adenomas. This would include MEC, AdCC, AciCC, intraductal carcinoma, SDC and, most importantly, carcinosarcoma.
Second, subclassifying the malignant tumors into low-grade and high-grade is mandatory. With this adding up (perhaps by inserting a new column for grading in each table, the discussion of epigenetic alternation in high-grade transformation should be heightened.
Third, tumor-to-tumor interaction and its role in posing diagnostic and prognostic challenges should be discussed adequately.
Fourth, it is insufficient to provide generic conclusions, such as "of diagnostic value" or "with therapeutic implications". Please define each!
Other minor issues:
The manuscript reads "There are 31 distinct primary neoplasms that can originate in the salivary glands, according to the most recent WHO classification guidelines in 2017" This is not true. Moreover, using the 2017's classification will add confusion for readers nowadays. Kindly use the 2022's classification. That said, kindly provide implications on taxonomical suggestions (e.g. should carcinosarcoma be considered a separate entity or an end-stage of carcinoma ex pleomorphic adenoma).
Using three acronyms for the same concept is confusing[Salivary gland tumors(SGTs), Salivary gland neoplasms(SGNs), and Salivary gland cancers (SGCs)]. I recommend using Salivary gland carcinomas (SGCs) given that the main focus is not given to sarcomatous transformation.
Author Response
RESPONSE TO REVIEWER 1
Thank you very much for your constructive comments. We have made the revision to the manuscript “The epigenesis of salivary glands carcinoma: from field cancerization to carcinogenesis’ as suggested as below;
- The introduction has been revised and shortened to include relevant points only.
- The benign tumour like pleomorphic adenoma and Warthin’s tumour are excluded from the manuscript and other points on these in the whole manuscript has been deleted. This is replaced by malignant tumour like carcinoma ex pleomorphic adenoma, acinic cell carcinoma, salivary duct carcinoma, adenoid cystic carcinoma and intraductal carcinoma.
- In surgical practice, the low-grade and high-grade tumours are only applicable to mucoepidermoid carcinoma. The other types of malignant tumours are behaving as of high-grade tumours. Thus, we don’t have any much information on high grade tumours of each type of malignant tumours to be included in the table.
- Tumour to tumour interactions and it’s roles in diagnosis and prognosis has been added to the revised manuscript. This is as below;
3.6 Tumour to Tumour interactions
Tumour to tumour interaction plays critical role in the carcinogenesis. Imperatively, there are multiple factors and proteins produced by the tumour and these tumoural markers create a complex interaction in the tumour ecosystem. The interaction of some of these markers in the tumour microenvironment has significant impact on the tumour biology and characteristics. For instance, the intratumoural lymphocytes has been associated with high risk of neck nodes metastasis and high-grade tumours in acinic cell carcinoma [129]. This is likely due to the presence of PDL1 expression in higher level in association with elevated immune cell infiltration of T and B cells. This underlies why the acinic cell carcinoma has been shown to have unfavourable prognosis and has higher risk of lymph node metastases.
Additionally, the process of cancer metastases is important as it influence the prognosis of patient with SGCs. The cancer cells infiltrate the lymphatic and blood vessels through the migration of extracellular matrix, where the main enzyme systems of MMPS is required, and this is located in the invadopodia of cancer cells. In adenoid cystic carcinoma, Lissencephaly 1 (L1S1) regulates the invadopodia formation and has been shown to associate with matrix metalloproteinases (MMPs) expression. Lissencephaly L1S1 is a microtubule associated protein which regulates the microtubules stability, and it can mitigate the metastatic potential of ACC through the invadopodia formation and ECM degradation [130]. Also, integrin linked kinase (ILK), play important role in ECM interactions, with presence of other cofactors such as growth factors and integrin, which regulates cells differentiation, migration and apoptosis. This has positive roles in tumour progression and transformation [131].
- 5. The statement “There are 31 distinct primary neoplasms that can originate in the salivary glands, according to the most recent WHO classification guidelines in 2017” has been omitted from the text. There is not much research data on carcinosarcoma as it is a very rare tumour, thus, do not able to include the details.
- The term "of diagnostic value" or "with therapeutic implications" has been revised to improve the meaning of sentences accordingly.
- The term ‘carcinomas’ has been used to replace the ‘neoplasms’ and ‘cancers’ within the whole manuscript, including the title.
Thank you very much.
Reviewer 2 Report
Salivary gland malignancies are rare, a diagnosis by pathologists is challenging and limited options for treatment are present. Additionally, a lot of sub-entities make diagnosis difficult.
The most important etiology for this cancer are epigenetic events. Until now, numerous pathways and epigenetic alteration has been identified.
The purpose of this review is therefore to provide a comprehensive overview regarding the role of epigenetic alterations as predictive, diagnostic biomarkers, and therapeutic targets in the management of SGCs.
In this review, the epigenetic mechanisms are generally described and in table 1 all published epigenetic markers associated with SGC are presented
In the next chapter Epigenetic alterations in salivary gland tumors are reviewed and in table 2 methylated genes contributing to the pathogenesis of SGCs are listed. In table 3 diagnostic and prognostic biomarkers in SGC are shown.
In summary, the authors have collected an immense amount of information on this important topic and presented it perfectly in text, tables, and images.
Author Response
REPONSE TO REVIEWER 2
Thank you very much for your positive comments on the manuscript. We are very delighted and will ensure the manuscript will be free of technical error.
Thank you.
Reviewer 3 Report
The presented review work aims to provide a comprehensive overview of the most up-to-date information regarding the role of epigenetic alterations as predictive, diagnostic biomarkers and therapeutic targets in the management of salivary gland cancers (SGCs). The authors conduct a thorough evaluation of the currently known evidence on the involvement of epigenetic processes in SGCs.
As known SGCs are a diverse collection of benign as well as malignant tumors with marked differences in biological activity, clinical presentation, and microscopic appearance. Although the etiology is varied, secondary radiation, oncogenic viruses as well as chromosomal rearrangements have all been linked to the formation of SGCs. Epigenetic modifications may also contribute to the genesis and progression of SGCs. Epigenetic modifications are any heritable changes in gene expression that are not caused by changes in DNA sequence. It is now widely accepted that epigenetics plays an important role in SGCs development. A basic epigenetic process that has been linked to a variety of pathological as well as physiological conditions including cancer formation, is DNA methylation. Transcriptional repression is caused by CpG islands hypermethylation at gene promoters, whereas hypomethylation causes overexpression of a gene. Epigenetic changes in SGCs have been identified, and they have been linked to the genesis, progression as well as prognosis of these neoplasms.
As the authors conclude, all of these molecular alterations might be exploited as prognostic molecular markers and could assist in refining the management of SGCs.
The review article is written and organized very well.
In my view, it can accept in its present form.
Author Response
RESPONSE TO REVIEWER 3
Thank you very much for your positive comments on the manuscript. We are very delighted and will ensure the manuscript will be free of technical error.
Thank you.
Reviewer 4 Report
This is a review of the epigenesis of salivary gland malignancy related to cancerization and carcinogenesis.
The overall manuscript is well constructed, and it is worth accepting. However, the authors should reconsider some points.
The authors summarized Table 1, but in what order are these references done? Tables 2 and 3 are the same.
In Table 2, references are almost the same. Please check it.
On page 16, "The epigenetic changes have been...on this genetic pathway" and page 3 overlap. The authors should avoid duplication.
On page 17, the hTERT gene was mentioned in the paragraph about cyclin-dependent kinase inhibitors (CKIs). Is hTERT related to CKIs? If they are not related, you should separate them.
There are two "3.4 DNA Methylation in acinic cell carcinoma."
In "DNA Methylation in acinic cell carcinoma," the contents of AdCC are mixed.
Pathological and clinical factors are discussed in new literature that has recently been published (Gutschenritter T, Machiorlatti M, Vesely S, Ahmad B, Razaq W, Razaq M. Outcomes and Prognostic Factors of Resected Salivary Gland Malignancies: Examining a Single Institution's 12-year Experience. Anticancer Res. 2017 Sep;37(9):5019-5025. and Nishida H, Kusaba T, Kawamura K, Oyama Y, Daa T. Histopathological Aspects of the Prognostic Factors for Salivary Gland Cancers. Cancers (Basel). 2023 Feb 15;15(4):1236).
The authors described epigenetic drugs in SGC, so you should add those effects.
Although the authors showed diagnostic biomarkers in Table 3, almost genes/elements were not used for actual pathological diagnosis by pathologists. Please correct it.
The authors should recheck References; for example, references 4 and 74 are the same.
Author Response
RESPONSE TO REVIEWER 4
Thank you very much for your constructive comments. We have made the revision as suggested as follows;
- The references for Table 1 are following the order of appearance in the text. The numbers are in line with the listed reference.
- Table 2 and 3 are the not exact the same. Table 2 is on the description of methylated genes contributing to the pathogenesis of SGCs, whereas Table 3 is on the diagnostic and prognostic biomarkers in SGC and has additional information in addition to the listed methylated genes.
- References for Table 2 has been revised accordingly.
- The overlap statement “The epigenetic changes have been...on this genetic pathway” has been omitted. The related reference has been revised.
- The paragraph on hTERT gene has been separated from the cyclin-dependent kinase inhibitors (CKIs) as suggested.
- The repeat of “3.4 DNA Methylation in acinic cell carcinoma” has been revised. The number of the following subheading has been revised.
- In "DNA Methylation in acinic cell carcinoma has been revised and separated from the contents of Adenoid cystic carcinoma AdCC
- The articles on the pathological and clinical factors as in the suggested article has been added and included in the reference list.
(74. Gutschenritter T, Machiorlatti M, Vesely S, Ahmad B, Razaq W, Razaq M. Outcomes and Prognostic Factors of Resected Salivary Gland Malignancies: Examining a Single Institution's 12-year Experience. Anticancer Res. 2017 Sep;37(9):5019-5025.)
(137. Nishida H, Kusaba T, Kawamura K, Oyama Y, Daa T. Histopathological Aspects of the Prognostic Factors for Salivary Gland Cancers. Cancers (Basel). 2023 Feb 15;15(4):1236).
- The effects of epigenetic drugs in SGC have been added as in the Table 4.
- The list in table 3 is the potential diagnostic biomarkers. So, yes, it is true that almost all genes were not used by the pathologist for pathological diagnosis, however, it may be utilized at the near future.
- The duplicated reference has been revised and omitted and replaced with related one.
Thank you very much.